# Karenina: Modeling the Complexity of Negative Emotions to Better Serve Industry Goals

**Moritz Sudhof**
Motive Software
moritz@motivesoftware.com

**Liam Croteau**
Motive Software
liam@motivesoftware.com

**Christopher Potts**
Stanford University
cgpotts@stanford.edu

## Abstract

Sentiment analysis systems are widely applied in industry, but standard formulations of the task (e.g., positive/negative/neutral classification) are often not well aligned with real-world goals. For instance, in customer support contexts, negative labels dominate due to the nature of the work, and different negative emotions call for different solutions. To help address this issue, we introduce Karenina, a labeled dataset of 25K consumer healthcare experience comments with labels that support standard sentiment distinctions but also allow for a breakdown into six negative emotions: confused, disappointed, frustrated, angry, stressed, and worried. Each text has 1–4 emotion labels, with over 90% of examples having at least 2 labels. We define strong baselines for this dataset, we seek to motivate a flexible approach to evaluation that takes into account the variable costs for different mistakes in different industrial contexts, and we report on some illustrative analyses using Karenina to understand customer experiences.

## 1   Introduction

Textual sentiment and emotion analysis are widely applied in industry, as tools to help firms manage their reputations and understand customer attitudes and experiences [13]. It's a mature market that is eager for new data, insights, and models from the scientific literature, and thus it presents an opportunity for researchers seeking to test their ideas in the real world. However, we find that the field of natural language processing (NLP) has mostly not been responsive to these needs. For example, of the hundreds of sentiment-related datasets available on Google Dataset Search, only a handful support more than positive/negative distinctions, whereas business leaders say things like "Sentiment is essential. It's even better when you can look at subtleties such as emotion and intensity" [14].

In this paper, we seek to help address this gap between research and practice by introducing Karenina, a labeled dataset of 25K consumer healthcare experience comments drawn from the Yelp Academic Dataset.[1] The name 'Karenina' comes from the famous opening line of Tolstoy's *Anna Karenina*: "Happy families are all alike; every unhappy family is unhappy in its own way". In many customer service and support contexts, most of the examples are negative, and the key questions turn on the nature of those negative emotions. Thus, Karenina has labels supporting a breakdown into six negative emotions (confused, disappointed, frustrated, angry, stressed, worried) in addition to standard sentiment distinctions. The taxonomy of emotions used here is based on our own experience doing

---

[1] The Karenina dataset is available at https://github.com/sudhof/karenina.

Table 1: Randomly selected short examples and full annotation count distributions from Karenina.

| Text | Angry | Confused | Disappointed | Frustrated | Neutral | Positive | Stressed | Worried |
|---|---|---|---|---|---|---|---|---|
| Asked no questions and kept filling out my chart. | 0 | 3 | 3 | 0 | 3 | 0 | 1 | 1 |
| Again, I assumed this was taken care of. | 0 | 1 | 3 | 3 | 2 | 0 | 1 | 3 |
| One time he made my neck worse! | 2 | 0 | 6 | 8 | 0 | 0 | 3 | 0 |
| He has raised the bar so high that I now am afraid I will never be happy with another dentist. | 0 | 0 | 0 | 0 | 4 | 6 | 2 | 3 |
| I came in with severe tooth pain from a missing filling. | 1 | 0 | 1 | 2 | 2 | 0 | 3 | 2 |

emotion analysis in industry. Table 1 provides a selection of (short) examples with labels inferred from the choices made by our annotators.

We anticipate that models trained on Karenina can be used to identify customers that need additional intervention or escalation, to measure the effectiveness of support services and motivate improvements, and to build more empathic chatbots, among other uses. To begin exploring this area, we fine-tune BERT models [9] on Karenina, taking two approaches: binary classifiers for each emotion, and a multi-class classifier trained on the full response distributions from our annotators. We find that the binary classifiers are more effective, but the multi-class ones are competitive and easier to build and maintain.

Karenina also leads us to reconsider standard assumptions about how to evaluate applied systems in this space. The emotions expressed in Karenina have complex relationships to each other. For instance, if a text has 'Disappointed' among its true labels, it is much less problematic for a system to guess 'Frustrated' than for it to guess 'Positive'. The first error might not even be noticed by someone using the system, whereas the second is embarrassing or worse. Similarly, 'Confused' and 'Stressed' have semantic dimensions that intuitively set them apart from the others. We would like to capture these distinctions in our evaluations, so we argue for using weighted per-category F1 scores to partially encode our goals and values in a familiar set of metrics. We present a method for inferring these weights based on Karenina itself.

## 2 Related Work

**Industry Applications and Needs**   There is an increasing need for organizations to understand and optimize people's experiences by leveraging unstructured data in the form of chat interactions, call transcripts, surveys, reviews, and other natural language sources. The COVID-19 pandemic has only accelerated these trends [30]. And investments in experience make business sense. For instance, according to a recent research report from Salesforce [25], 57% of customers have stopped buying from a company because a competitor provided a better experience. Readily available sentiment analysis tools purport to be able to address these needs, and yet companies still struggle to understand and act on textual experience data. For instance, according to Forrester [11], 71% of companies don't have the ability to aggregate and mine customer feedback across surveys and other sources, and 79% of companies are unable to quantify the business impact of customer experience issues.

**Sentiment Analysis Benchmarks**   Sentiment analysis was one of the first benchmarks for natural language understanding systems [20; 21; 29], and it remains a very common task today. Pang and Lee [20] introduced one of the first public datasets in this space, based in sentences from movie reviews, and that dataset subsequently became the basis for the Stanford Sentiment Treebank [26]. The largest datasets tend to be based in consumer product reviews [15; 17; 19]. Social media posts are

another common source. Potts et al. [24] released a dataset combining naturally occurring examples with those generated by crowdworkers seeking to fool a top-performing model [16]. In industrial contexts, models trained on these benchmarks can provide only a high-level picture. On seeing that a percentage of customer experiences are negative, businesses will want to know more about the nature of those experiences, and sentiment alone does not offer that deeper layer of insights.

**Emotion Analysis Benchmarks**  Emotion analysis benchmarks have the potential to help us reach those deeper insights, since they often support further distinctions beyond affective valence. Strapparava and Mihalcea [27] released a dataset of roughly 1500 news headlines rated along six different emotions (similar to [10]) as well as a valence (positive/negative) score. The ratings were assigned by expert annotators. This is one of the first multidimensional emotion datasets; while its small size limits its usefulness as a resource for training modern systems, it still has value as a source for system assessment. Aman and Szpakowicz [2] employ similar methods, but with a focus on social media and explicit alignment with the categories of Ekman [10].

A number of emotion benchmarks offer noisy labels derived from Twitter hashtags and other kinds of metadata [1; 18; 31]. These datasets are more wide-ranging in their emotional categories but correspondingly less controlled in terms of the consistency with which specific labels are used.

The closest effort to our own is the GoEmotions dataset of hand-labeled Reddit comments, due to Demszky et al. [8]. GoEmotions uses 28 categories, with labels derived from a crowdsourcing effort. Individual texts can be labeled with multiple labels, and crowdworkers were free to assign as many labels as they wished. We see Karenina as building directly on the insights of GoEmotions. The three major differences are as follows:

1. The GoEmotions task structure resulted in relatively few examples with more than one emotion label (17%), whereas we developed a methodology that led to a much higher rate of examples with multiple labels.

2. For GoEmotions, only three raters were assigned to each example, and if two agreed on an emotion label, that emotion was considered the single gold label for the text. Only in the case where none of the three raters agreed did two additional raters get assigned, leading to the possibility of multiple labels. In contrast, our task was designed to generate not a single emotion label for a text but a distribution across emotions. For this reason, we assigned substantially more raters per text than GoEmotions (after filtering annotations for quality control, the vast majority of examples still had at least 6 unique annotators).

3. We apply a different taxonomy of negative emotions that is better suited to our domain (healthcare consumer feedback). For instance, Remorse, Grief, and Embarrassment are prevalent in Reddit forum comments but not in our data, whereas our additional category Worried is vital for understanding customer feedback in our space. Similar modifications might be called for in other business contexts.

## 3   Karenina

We now describe the construction of Karenina, emphasizing how we sought to obtain relevant high-quality examples expressing multiple emotions (Sections 3.1–3.5) and how we designed our train/dev/test split (Section 3.7).

### 3.1   Constructing the Dataset

The sentences in Karenina are extracted from healthcare reviews in the Yelp Academic Dataset,[2] which contains over 8.6 million reviews for 1.6 million businesses, spanning many industries. We chose this dataset as the basis for our corpus because it is large, publicly available, and has extensive metadata that could support a wide range of analyses beyond the classification tasks we explore here.

For this paper and dataset, we restrict our focus to reviews of healthcare businesses such as hospitals, emergency centers, medical clinics, dentists, and other medical services. The healthcare domain exhibits characteristics that make it an interesting focus for this paper: (1) healthcare is a large industry where understanding and improving patient experiences is an ongoing struggle, (2) extensive

---

[2]`https://www.yelp.com/dataset`

patient experience data is collected due to regulations, and (3) patients interact with healthcare providers during particularly significant or vulnerable moments in their lives, leading to a high level of texts expressing nuanced negative emotions.

After filtering out non-healthcare Yelp reviews, we were left with 161,480 reviews from 8,314 unique businesses, of which we sampled 50,000 reviews. We then applied NLTK's sentence tokenizer [4] to tokenize reviews into a total of 427,608 sentences. We filtered out sentences that were fewer than 10 or more than 200 characters, and we removed duplicate sentences. This resulted in 396,988 sentences.

Randomly sampling from our collection of sentences would lead to an under-representation of the rare but vital emotions that we wish to target, like 'Angry' and 'Confused' and an over-representation of broad classes such as 'Positive'. To ensure our dataset includes relevant data for the study of negative emotions, we sampled with the aid of simple heuristic classifiers, primarily to try to reduce the number of positive or neutral texts in our corpus. Following Potts et al. [24], we also favored sentences where the classifier predictions clashed with the review-level rating, with the expectation that this would surface hard examples, and we extended this to cases where the simple classifiers' predictions seemed to conflict (e.g., "worried" and "grateful"). We emphasize that, outside of this sampling, the labels from these classifiers are not used in Karenina; all labels are assigned by humans (Section 3.3).

## 3.2 Emotion Taxonomy

Emotion taxonomies in NLP are often grounded in psychological research, as we see in the brief review of existing datasets in sec:related. In contrast, we motivate our emotion taxonomy based on industry experiences, seeking to help provide actionable insights.

1. **Frustration**: Frustration indicates distress or annoyance at being unable to achieve a goal. As such, it is a good indicator of customer effort, a common metric organizations try to minimize. Frustration is additionally important to track because it creates a need to respond in individuals, which often manifests as aggression or hostility, so frustration can be an "early warning sign" of negative customer reactions like churn [3; 5].

2. **Disappointment**: Disappointment signals that an expectation has not been met. It is semantically similar to frustration, but it has less to do with effort and other outward indicators and more about identifying gaps in expectations versus reality.

3. **Confusion**: Confusion is the most actionable emotion in our taxonomy, since it provides key insights to how a business could immediately clarify their communication with customers in order to increase satisfaction and trust.

4. **Worry**: Worry is a forward-looking emotion, serving as a preemptive signal prior to a negative event happening, allowing for the opportunity to intervene early to improve outcomes.

5. **Stress**: Stress indicates an active state of anxiety, tension, or inability to cope due to feeling overwhelmed. Although stress is semantically similar to worry and frustration, it is important to isolate and track independently because it can have adverse and long-term physiological and cognitive effects.

6. **Anger**: Anger is a very strong negative emotion that signals a potentially serious problem because of the degree of emotional activation observed.

The exact descriptions for each emotion provided to our annotators can be found in the sample task in Appendix B.

## 3.3 Annotation

One of our central goals is to capture the fact that individual texts can express multiple emotions. This goal is motivated by psychological studies on emotion that conclude that emotion is characterized not by discrete states but gradients between emotion categories, and that emotional states fundamentally occupy a complex, high-dimensional categorical space, as described by [23], [7], [6]. The question then arises of how best to obtain annotations that reflect this multidimensionality.

In pilot work, we explored a binary formulation: for each example and each emotion, workers were asked whether the example expressed that emotion, with response choices 'yes', 'no', or 'borderline'.

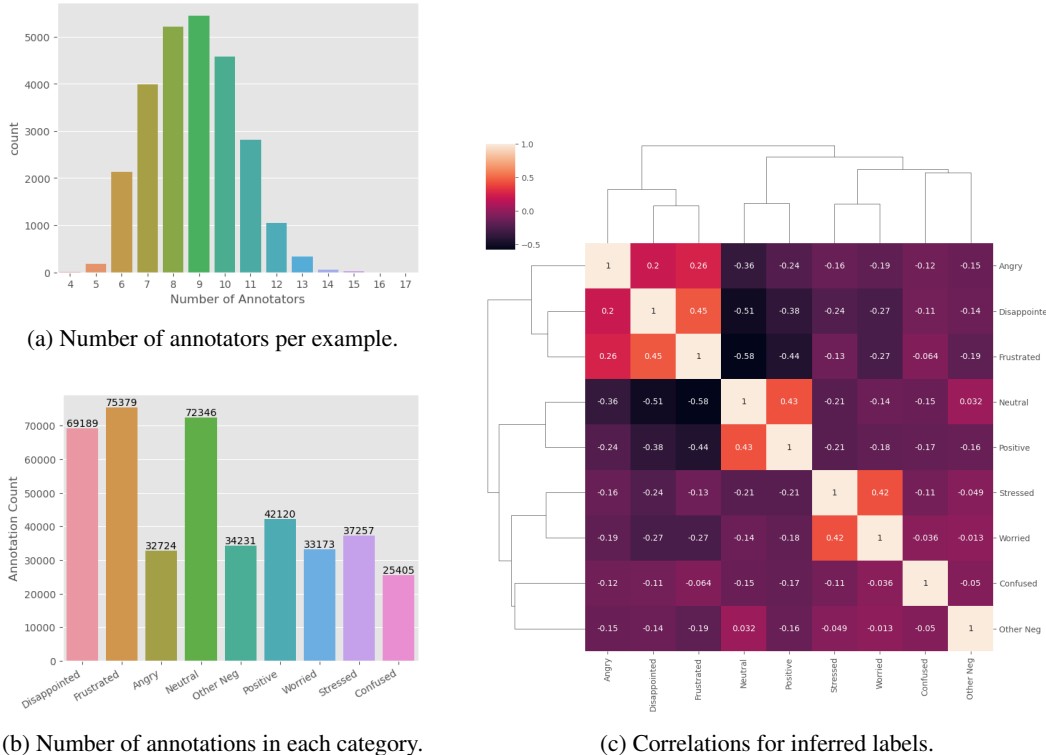

(a) Number of annotators per example.

(b) Number of annotations in each category.

(c) Correlations for inferred labels.

Figure 1: Dataset summary.

In theory, this would give us a very clear list of emotions present in an example. However, we found that this gave very noisy results; many annotators would always choose 'yes', perhaps as a strategy of erring on the side of caution, or perhaps because the task demands were too low. In addition, the costs of this approach are high, since each example/label pair becomes a separate task.

As noted above, Demszky et al. [8] allowed annotators to choose any number of labels, with the opposite result from our binary task: few examples ended up with multiple labels. This led us to a 'Top 2' labeling approach, in which annotators were asked to choose the top two labels (although if annotators felt only one label was appropriate, we did not prevent them from only selecting one). Requesting a second label helped to avoid a situation in which only the most general and prevalent emotions (Frustrated, Disappointed) were chosen. We piloted a 'Top 1' variant and found it to be good as well, but it yielded fewer secondary and tertiary emotions.

Appendix B provides additional details on our exclusion criteria for annotations.

### 3.4 Dataset Summary

Figure 1 provides a high-level picture of our dataset. The number of annotators who worked on each example ranges from 4 to 17, and the vast majority of examples had at least 6 annotators (Figure 1a). In Figure 1b, we summarize the raw annotations provided across the entire dataset. It matches with our prior expectations about the distribution of these emotions in texts like ours, and it also shows that the entire label set was used extensively.

### 3.5 Emotion Correlation

To better understand and motivate how to evaluate a model trained on this dataset, we analyzed the relationships between emotions in our taxonomy. Let $N$ be the number of examples in our dataset. We obtain $N$ dimensional vectors for each emotion $e$ by calculating, for each example, the proportion of raters' votes that were $e$. We then calculated the Pearson correlation between each pair of emotions and ran hierarchical clustering over the emotions using the correlation values as the distance between

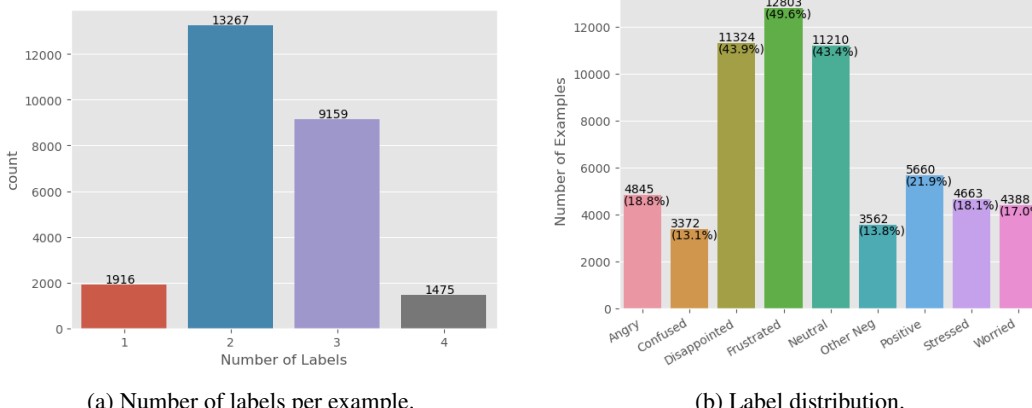

(a) Number of labels per example.

(b) Label distribution.

Figure 2: Summary of inferred labels.

Table 2: Dataset splits. 'Anns' is the number of individual annotation choices for each category, and 'Inferred' is our inferred label according to the method in Section 3.6.

|  | Train | | Dev | | Test | |
|  | Anns | Inferred | Anns | Inferred | Anns | Inferred |
| --- | --- | --- | --- | --- | --- | --- |
| Angry | 22,334 | 3,259 | 5,431 | 798 | 5,470 | 788 |
| Confused | 17,479 | 2,281 | 4,258 | 568 | 4,119 | 523 |
| Disappointed | 47,155 | 7,566 | 11,670 | 1,910 | 11,563 | 1,848 |
| Frustrated | 51,446 | 8,565 | 12,544 | 2,114 | 12,612 | 2,124 |
| Neutral | 49,927 | 7,597 | 11,713 | 1,774 | 12,106 | 1,838 |
| Positive | 29,060 | 3,831 | 6,889 | 911 | 6,993 | 917 |
| Stressed | 25,351 | 3,096 | 6,348 | 769 | 6,265 | 798 |
| Worried | 22,792 | 2,958 | 5,602 | 723 | 5,424 | 707 |
| Other Negative Emotion | 23,804 | 2,466 | 5,561 | 539 | 5,557 | 557 |
| Total | 289,348 | 41,619 | 70,016 | 10,106 | 70,109 | 10,100 |

emotions. The result of both these calculations are represented in the heatmap in Figure 1c. We can clearly see that some emotions are much more related to each other than others. For example, the higher intensity emotions of Frustration, Disappointment, and Anger are all highly correlated and form a subgroup. In contrast, each negative emotion has a negative correlation to the Positive label, as expected. In addition, the category Confused seems to be highly dissimilar from all the others.

## 3.6 Label Inference

There are many ways in which one might go about inferring labels from our annotation distributions. In our analysis, we stipulated that a label was present for an example if at least 33% of annotators chose it. As a result, each label for any example in the dataset has at minimum 3 annotators that agreed on a label, with 97% of examples having at least 4 annotations per label. Figure 2a gives the breakdown of the number of labels per example, and Figure 2b shows that the resulting dataset is truly a multi-label dataset, with the vast majority of examples having at least two inferred labels.

## 3.7 Dataset Splits

Table 2 summarizes our dataset splits. To create our splits, we first removed samples where the top inferred emotion was Other Negative Emotion, since we do not include this emotion in validation, and then randomly sampled 33% of texts for validation. This resulted in 8,433 validation texts, which were randomly split into dev and test sets of size 4,216 and 4,217, respectively. The resulting training set has 17,383 samples.

# 4   Modeling

We now report on experiments designed both to help validate Karenina and to show how it can be used to connect with real-world concerns for emotion analysis in industry. For this work, we excluded the 'Other Negative Emotion' class due to its rarity and lack of coherent semantics relative to the other labels. Additional implementation and optimization details are in Appendix C.

## 4.1   Metrics

We assess our models via F1 scores, on a per-class basis, and we macro-average those scores for a summary score, reflecting our view that all the categories are important, even the very small ones.

However, as we noted in Section 1, F1 scores by themselves do not reflect the fact that different mistakes have different costs in a real-world deployment; assigning the label 'Disappointed' to a text that should be labeled 'Frustrated' according to the gold data might not even register as a mistake with a user, but assigning 'Positive' could be quite problematic.

These considerations motivate us to consider a weighted scoring metric that we call the Accepted Collateral Score (AC-score). Given a pair $(y, \hat{y})$, where $y$ is the true label and $\hat{y}$ the predicted label, $AC(y, \hat{y}) \geq 0$. We assume that the $AC(y, \hat{y}) = 0$ where $y = \hat{y}$. If $AC(y, \hat{y}) = 1$ wherever $y \neq \hat{y}$, then AC scores reduce to standard error assessments.

Our intention is for AC-scores to represent when our model is still 'rightish' compared to being completely wrong, so we associate more severe mistakes with larger values. Once AC scores have been calculated, we can report F1 scores as usual (and can in principle consider different weightings of precision and recall beyond F1, as a separate way of expressing different values we might have).

AC-scores could derive from a number of sources: human evaluations of the costs of different mistakes, business costs that can be indirectly linked to model mistakes, and so forth. In this paper, we seek to leverage Karenina to directly estimate such scores. To do this, we used the label correlation matrix in Figure 1c and then re-scaled each column to be between $-1$ and $0$ and multiplied the result by $-1$ to get scores between $0$ and $1$. This gives us a matrix that follows the previous rules outlined, where scores closer to $0$ signify more leniency between label pairs for scoring.

To calculate $F1_{AC}$ for an experiment given a confusion matrix, the confusion matrix is multiplied element-wise with these AC-scores in order to achieve a re-weighting of individual errors, after which F1 is computed as normal on the basis of the re-weighted confusion matrix.

Finally, because we have full response distributions for all examples, we can consider our examples to be labeled with probability distributions over our categories, obtained by normalizing these response distributions. This allows us to use average KL-divergence as a secondary metric.

## 4.2   Models

**Random Classifiers**   As baselines, we use random classifiers that make predictions proportional to the distribution of labels in the training data. We use these to help quantify the extent to which our models are able to get traction on Karenina. To report average KL-divergence for these models, we aggregate each model's predictions and softmax across all labels.

**Binary Classifiers**   We fit separate binary classifiers for each emotion label in Karenina, treating them as separate tasks. These models begin with pretrained `bert-base-uncased` parameters with a dense layer added on top for the purposes of fine-tuning for classification. To calculate a KL-divergence score for these models, we gather all positive class logits from each binary model and apply softmax in order to create a probability distribution over all the labels.

**Multi-class Classifiers**   Our multi-class classifier models our dataset in a unified way. Similar to the binary models, we begin with pretrained `bert-base-uncased` parameters with a dense layer added on top for the purposes of fine-tuning for classification. Unlike our binary classifiers, our multi-class classifiers do not make use of our inferred labels during training. Rather, they embrace the full set of annotations we received for each example: in the training data, each example is labeled with a full probability distribution defined by normalizing the response counts. We can then assess these models using KL-divergence as well as standard F1-style metrics, with and without AC-score

Table 3: Main results. The binary model achieves KL-divergence scores of 0.31 and 0.32 on dev and test respective, whereas the multi-class achieves only 0.49 on both splits.

| | Random (Binary) | | Binary | | Random (Multi-class) | | | | Multi-class | | | |
| | Dev | Test | Dev | Test | Dev | | Test | | Dev | | Test | |
| | F1 | F1 | F1 | F1 | F1 | F1$_{AC}$ | F1 | F1$_{AC}$ | F1 | F1$_{AC}$ | F1 | F1$_{AC}$ |
|---|---|---|---|---|---|---|---|---|---|---|---|---|
| Angry | 17.3 | 17.1 | **75.2** | **75.8** | 4.9 | 6.4 | 7.7 | 10.1 | 50.4 | 64.2 | 53.9 | 67.0 |
| Confused | 11.9 | 12.2 | **71.7** | **69.2** | 8.5 | 9.9 | 7.9 | 9.1 | 63.8 | 68.1 | 60.3 | 64.6 |
| Disappointed | 45.0 | 44.8 | **72.6** | **70.8** | 17.2 | 21.4 | 19.1 | 23.8 | 43.0 | 52.8 | 43.1 | 53.1 |
| Frustrated | 50.0 | 49.5 | **72.4** | **73.3** | 20.9 | 25.7 | 22.9 | 28.1 | 35.7 | 45.7 | 33.8 | 43.1 |
| Neutral | 42.0 | 43.9 | **71.7** | **72.1** | 20.2 | 23.0 | 22.6 | 25.3 | 50.0 | 55.6 | 50.7 | 56.1 |
| Positive | 21.8 | 20.7 | **62.3** | **60.5** | 12.5 | 15.5 | 12.0 | 14.7 | 51.6 | 61.8 | 50.1 | 59.6 |
| Stressed | 18.0 | 17.7 | **55.1** | **59.4** | 6.4 | 7.6 | 5.0 | 5.9 | 38.7 | 46.9 | 37.3 | 45.9 |
| Worried | 17.1 | 14.2 | **60.0** | **62.4** | 6.5 | 7.6 | 9.2 | 10.8 | 46.4 | 54.1 | 47.1 | 55.0 |
| Macro Avg. | 27.9 | 27.5 | **67.6** | **67.9** | 12.1 | 14.6 | 13.3 | 15.97 | 47.5 | 56.2 | 47.0 | 55.6 |

reweighting applied. In order to calculate F1-style metrics using the multi-class model, we transform our dev and test splits to gold label datasets where the gold label for an example is the emotion label with the highest response count. In the case of a tie for highest response count we duplicate the example in the dev and test set with each emotion label that tied for the highest response count as the gold label. We did this instead of randomly sampling from labels tied for highest response count for ease of reproducibility and to better reflect how the AC-score can be used to better understand results with complex relationships between labels.

### 4.3 Results and Discussion

Table 3 summarizes our results. We find that our binary and multi-class models are substantially better than the random ones, overall and for every individual category, which supports our contention that Karenina presents a coherent and interesting learning target.

We also see that the binary models yield higher F1 scores on a per label basis, and in aggregate, compared to the multi-class classifier. The binary models are also better in terms of average KL-divergence. This suggests that the binary models are better both at determining whether an emotion is present and at modeling the full emotion distributions. The binary models are substantially more expensive to fit and maintain than a single multi-class one, but the costs might be worth it if the goal is raw predictive accuracy.

The results also suggest that AC-scores are useful in the context of real-world deployments. For example, the F1$_{AC}$ scores for emotions like 'Neutral', 'Angry', and 'Stressed' all see a large boost in their scores relative to the F1 scores, which shows, in effect, that the model isn't as wrong as the the normal F1 score make them out to be.

## 5 Empirical Analyses

To illustrate how models built from Karenina might be used in industry, we use the multi-class model from Section 4 as a device for understanding the broader dataset of 161,480 Yelp healthcare reviews. We show that our model not only predicts review ratings but that it can also be applied to specific businesses to suggest concrete plans for improvement.

### 5.1 Relationship between Emotions and Review Ratings

Although our emotion model does not incorporate review ratings in training, there are strong relationships between the emotions expressed in reviews and the ratings supplied by customers. To quantify this, we measured the mutual information between each emotion and the ratings. All emotions demonstrate a high degree of mutual information with ratings except for Worried, which has an intuitive explanation: reviews from satisfied customers still commonly contain signals about the customer's emotional state before a medical visit (e.g., "I was so afraid to go to the dentist") or reference the underlying medical condition that motivated the care experience to begin with (e.g.,

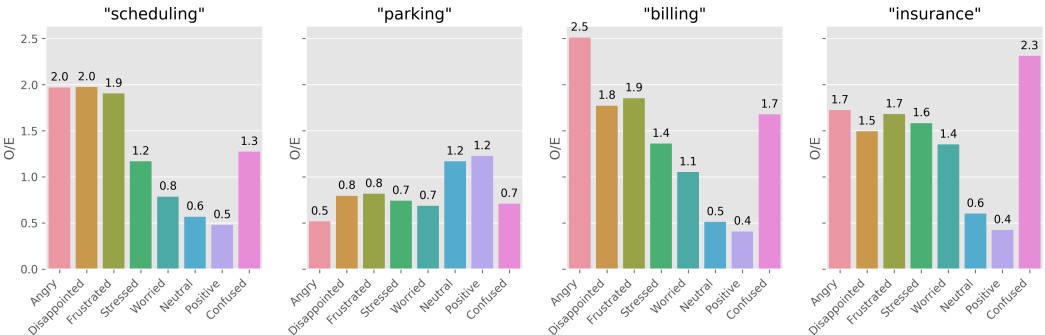

Figure 3: Observed over expected profiles for sample non-care-related key terms.

"The biopsy showed presence of cancer"). Appendix D provides additional examples, figures, and analyses to further characterize the relationship between review ratings and emotions.

## 5.2 Prioritizing Experience Improvements based on Emotions

More important to organizations than predicting ratings based on unstructured feedback is understanding what they can do to improve them. A sentence-level annotation of reviews for emotion can be immediately applied to surface concrete recommendations for improvement. To illustrate this application of our model, we conducted a study of the reviews for a single business, chosen at random from all businesses in the overall dataset that had more than 3,000 total reviews.

To highlight how simple methods can be effective when used in conjunction with a Karenina model, we adopt a keyword-based approach. To preprocess our data, we score all sentences for top emotion and tokenize each sentence into words. We filter out stopwords and count word occurrences to identify the most frequent words used in reviews of the selected business. These words represent common topics relevant to the customer experience of patients, from terms about health issues and treatment ("pregnancy", "obgyn", "c-section") to terms about staff and frontline service ("doctor", "nurse", "receptionist"), to terms about other aspects of the care experience, from scheduling and arrival to payment and billing ("parking", "waiting", "insurance").

By looking at the "emotional profile" of different frequent terms, we can isolate specific aspects of the customer experience that need improvement. For each term, we plot the observed over expected ratios for each emotion. We compute the expected score for an emotion $e$ by averaging the scores for emotion $e$ in all sentences in reviews for the target business, and we compute the observed score by averaging the scores for emotion $e$ in all sentences containing the term.

Figure 3 plots the results for a selection of terms relating to aspects of the patient experience unrelated to the quality of medical treatment. This analysis quickly identifies trouble spots. For example, for the selected business, "scheduling" can lead to high levels of disappointment, anger, and frustration, "parking" is not a problem, "billing" angers customers, and "insurance" is confusing. This analysis not only surfaces the key experience aspects companies should invest in but also helps companies understand what to do about it: knowing that "insurance" confuses customers suggests that investing in more information and transparency in the insurance process would go along way to alleviating the issue.

Appendix E provides a comparable analysis for staff-related terms, which again yields actionable insights about how our target business might improve its patients' experiences.

## 6 Impact Statement

**Research Limitations** The inherent limitations of our research stem from the defining features of our corpus Karenina; in focusing on the healthcare-oriented reviews and restricting attention to the emotional dimensions given in Section 3.2, we limit the domain in which our dataset can be usefully and responsibly applied. In turn, the major risks associated with our work come from people using Karenina models outside this domain. For example, the language of performance reviews might

be sufficiently different from that of healthcare reviews as to make Karenina models systematically biased in that domain. Similarly, the fact that we measure just the emotions in Section 3.2 means that we might miss, or mis-analyze, important signals that do not naturally fall into those categories.

**Societal Impact**   The research limitations we identified above are directly related to potential negative societal impacts of our work. It is also worth articulating more general issues that can arise for any kind of emotional detection technology. Even when functioning as intended, emotion detection technology could clash with the expectations of users and have unintended consequences for specific groups, especially minority groups. To some extent, these concerns are pervasive for AI technologies, but emotion analysis can have special weight.

## 7   Conclusion

We introduced Karenina, a dataset of sentences from consumer healthcare reviews with basic sentiment labels as well as labels for six negative emotions that, taken together, can help businesses address critical issues related to customer experiences in this domain. Karenina is multi-label dataset, with over 90% of examples having multiple labels, and our classifier experiments show that it is a coherent and challenging task. We also argued that models in this space should be assessed according to a weighted version of standard F1 metrics, to better capture the real-world costs of specific kinds of mistake. We hope that this new resource and associated set of results help to bring the field of emotion analysis in NLP closer to what is needed for industrial applications.

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

# Supplementary Materials

## A  Dataset Details

Karenina is available at `https://github.com/sudhof/karenina` in JSONL format, and is released under a Creative Commons 4.0 International license.[3] Appendix F provides a Datasheet [12] for it. Our corpus release includes only short text snippets from the public Yelp Academic Dataset. In keeping with the terms of the Yelp Academic Dataset, we advise that our dataset should be used only for research purposes. The authors bear all responsibility in case of violation of rights.

## B  Labeling Task

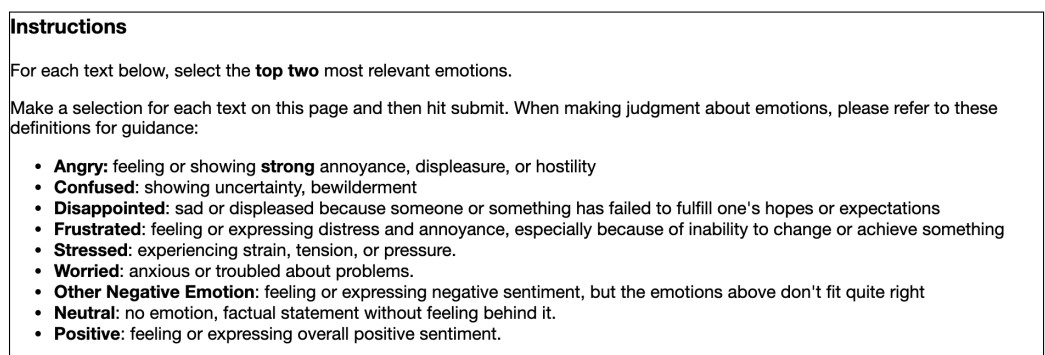

(a) Instructions provided to raters.

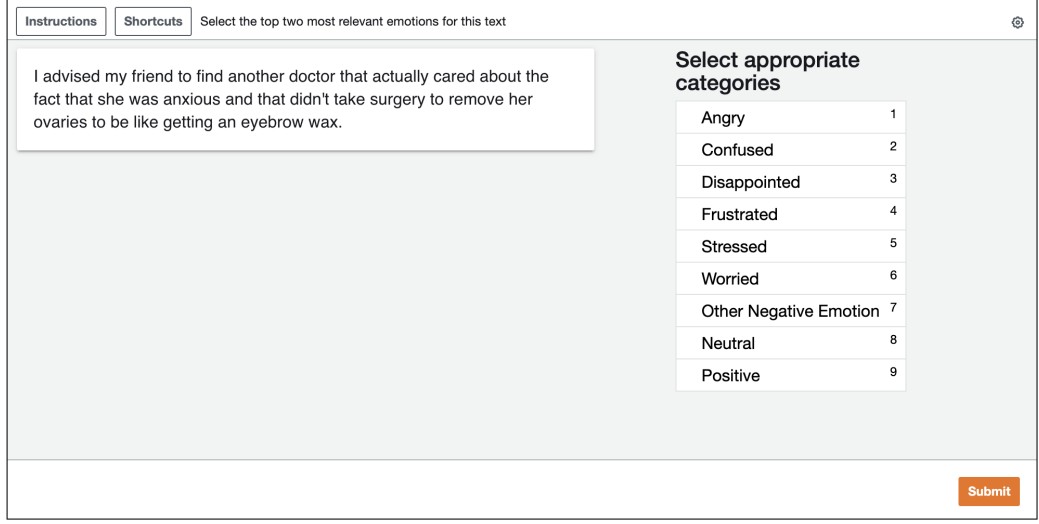

(b) Sample annotation interface for a single example.

Figure 4: Mechanical Turk annotation interface.

### B.1  Task Set-up

Figure 4 shows the interface for the annotation task used. Each Human Interface Task (HIT) included instructions that defined each emotion as well as ten sentences to be annotated. Workers were paid

---

US$0.50 per HIT, and all workers were paid for all their work, regardless of whether we retained their labels. Examples were uploaded to Amazon's Mechanical Turk in batches of around 5K examples.

## B.2 Exclusion Criteria

We filtered the annotations according to a number of criteria. First, all annotators that uniformly answered the same thing for each example were filtered out. Second, any annotator that spent less than 100 seconds on a batch of examples was filtered out. Third, we labeled 1000 example texts in-house and filtered out any annotator that got at least two of the gold labeled positive examples wrong. Where this filtering led to an example having too few annotations, we had it relabeled by a pool of annotators that had previously done high quality work on our task.

To remove workers from our pool, we used a method of 'unqualifying', as described in [28]. This method does no reputational damage to workers but allows us to disqualify them from participating in future rounds. This iterative approach allowed us to gather annotations from a broad set of workers while continuously monitoring and guiding annotation quality. We encouraged a broad range of interpretations and participation in the annotation task by having at least 9 unique workers annotate each sample text, and, even after quality filtering, the vast majority of our examples have at least 6 annotators. While we think our method mainly increased label quality, we recognize that it can introduce unwanted biases, and we provide further commentary on the dataset's potential biases in 6.

## B.3 Payment

Accurately estimating our hourly wage is challenging due to the fact that we allow workers to start multiple tasks and complete them on their own schedule. This leads to a highly skewed distribution of time-taken, with a few workers seeming to take hours to do a few labeling tasks.

Of all our HITs, 48% had a reported work time greater than 500 seconds and 30 had a work time greater than 1000 seconds. The vast majority of these HITs likely represent instances of workers keeping their browser window open while attending to other matters and therefore cannot be viewed as reliable representations of true work time.

Even despite these factors, if hourly wages are calculated on a by-worker basis (rather than a by-HIT basis, as Mechanical Turk does by default), the average hourly wage for workers is US $8.15. (Federal minimum wage in the US is $7.25.)

Furthermore, although it is not possible to know which countries workers are located in, our HITs were open to workers from multiple countries, most of which have a cost of living and minimum wage significantly below US levels, and it is likely that a not insignificant portion of our workers are located in these countries.

We believe in paying strong rewards for HITs. In our own internal tests, average time per HIT for our task was 69 seconds. Given our familiarity and experience in this space, we assume we are faster at tagging for emotion than the average crowdworker. Therefore, for a conservative estimate of HIT completion time for crowd workers, we assumed that we were 3x faster than an average worker, leading to an estimated worker completion time of 207 seconds per HIT. With a reward of US $0.50 per HIT, our (conservative) estimated hourly wage was US $8.70. Our final estimate of $8.15 is below this target, but still well above the US federal minimum wage.

The total amount spent on participant compensation was $26,020.20.

## C  Model Details

Our Random models are implemented using scikit-learn's DummyClassifier with the 'stratified' guessing strategy [22].

All of our non-Random models begin with `bert-base-uncased` parameters from the HuggingFace library [32] with a dense layer added on top for the purposes of fine-tuning for classification. We explored learning rates in the range $1e-4$ to $1e-5$. Throughout, we use the Adam optimizer, a batch size of 16, and a maximum sequence length of 32. We used the early stopping function from keras on the dev set loss on each epoch with a patience of 4 to decide when to stop training in the hyper parameter tuning process. We report the models that had the best macro F1 average scores.

Table 4: Model learning rates selected by hyperparameter search.

| Model | Learning Rate |
|---|---|
| Binary(Angry) | $9e{-}6$ |
| Binary(Confused) | $9e{-}6$ |
| Binary(Disappointed) | $9e{-}6$ |
| Binary(Frustrated) | $1e{-}5$ |
| Binary(Neutral) | $1e{-}5$ |
| Binary(Positive) | $9e{-}6$ |
| Binary(Stressed) | $1e{-}5$ |
| Binary(Worried) | $2e{-}5$ |
| Multi-class classifier | $1e{-}5$ |

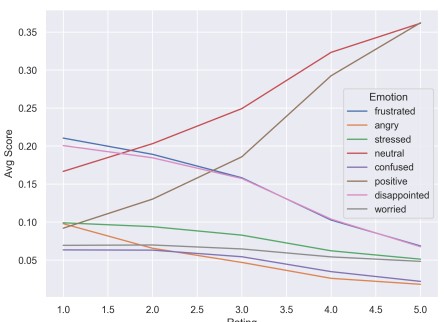

(a) Average emotion scores by review rating.

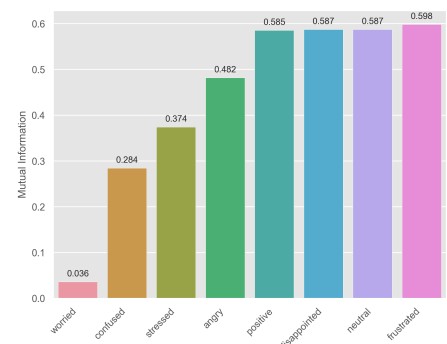

(b) Mutual information between each emotion and review ratings.

Figure 5: Relationship between emotions and review ratings.

For our binary classifiers, we conducted hyperparameter tuning on a restricted hyperparameter space based on our results for the multi-class classifier. For the binary models, we only considered learning rates in the range $5e{-}5$ to $1e{-}5$.

The final learning rates used for the reported models can be found in Table 4.

All models were trained in Google Cloud's AI notebooks environment with access to $4$ CPUs, a total of $15$ GB of RAM, and $1$ NVIDIA Tesla T4 GPU.

# D   Additional Connections Between Emotions and Review Ratings

Section 5.1 briefly studied the relationship between review-level ratings and emotions. Figure 5b shows the full set of relationships, and figure 5b provides all of the mutual information values.

Worried is the only emotion that is only weakly related to the ratings, which we trace to the complexity of this emotion in healthcare contexts. Table 5 provides randomly sampled examples to show that the Worried emotion is extremely diverse in terms of its causes, which helps explain why it has relatively little correlation with overall star ratings as compared to the other categories.

We also conducted an experiment where we fit a linear regression model to predict ratings based on emotion scores. We train on 80% of the overall reviews and test on the remaining 20%. This yields R-squared values of 0.809 on the training set and 0.810 for the test set. Our ability to predict ratings based on predicted emotion scores demonstrates that our model is capable of capturing distinctions in experience feedback that meaningfully predict a customer's overall evaluation of the experience.

Table 5: Randomly selected "Worried" sentences from 5-star reviews.

| Sentence | Rating | Predicted Emotion |
|---|---|---|
| I was obviously nervous about getting eye surgery. | 5 | Worried |
| All these one star reviews had me worried, but not as worried as I was about getting my husband some help ASAP. | 5 | Worried |
| My PSA showed abnormalities. | 5 | Worried |
| I had huge fears regarding how bad I had let my teeth go over the span of 10 years. | 5 | Worried |
| Years after a very high risk first pregnancy (I had severe pre-eclampsia and diabetes) I wasn't even sure I wanted to get pregnant again. | 5 | Worried |

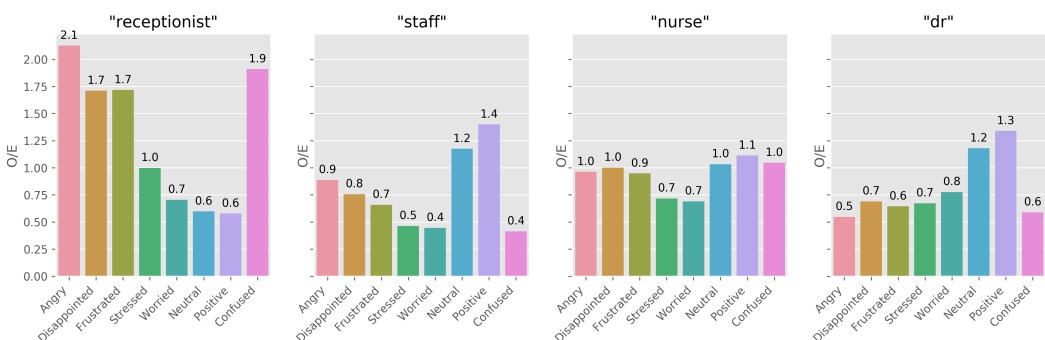

Figure 6: Observed over expected profiles for sample care-related key terms.

# E  Keyword-Analysis of Care-Related Terms

Section 5.2 reported on a keyword-based analysis focusing on non-care-related terms. Figure 6 provides a comparable analysis for terms that relate more directly to care. We see that customers are generally favorable of the frontline service they receive (see results for "staff" and "dr"), but mentions of the "receptionist" contain comparatively high levels of anger and confusion, suggesting that additional training or staffing of the front desk would improve overall experience.

# F  Datasheet

## F.1  Motivation

### F.1.1  For what purpose was the dataset created?

Karenina was created to facilitate the development of emotion analysis systems that can accurately capture a range of negative emotions that are indicative of particular customer experiences.

### F.1.2  Who created the dataset and on behalf of which entity?

The dataset was created by Moritz Sudhof (Motive Software), Liam Croteau (Motive Software), and Christopher Potts (Stanford University). All members of the team were functioning as independent researchers within their organizations.

### F.1.3  Who funded the creation of the dataset?

The effort was funded by Motive Software.

### F.2  Composition

#### F.2.1  What do the instances that comprise the dataset represent?

The instances are English-language sentences with labels and additional metadata. These sentences are records of acts of linguistic communication involving product and service evaluations in the space of consumer healthcare.

#### F.2.2  How many instances are there in total?

Version 1 (the current version) has 25,817 instances.

#### F.2.3  Does the dataset contain all possible instances or is it a sample (not necessarily random) of instances from a larger set?

The dataset contains a sample of sentences from the Yelp Academic Dataset, as described in Section 3.1 of the paper.

#### F.2.4  What data does each instance consist of?

The dataset is released in the JSON lines (JSONL) format. The README.md file in our project Github repository documents the dataset format.

#### F.2.5  Is there a label or target associated with each instance?

Yes, examples in Karenina have 1–4 labels drawn from the set 'Confused', 'Disappointed', 'Frustrated', 'Angry', 'Stressed', 'Worried', 'Neutral', 'Positive', and 'Other Negative Emotion'. In addition, each example includes the full response distribution from our crowdsourcing effort, with anonymized worker ids. Each example in the dataset also has a number of metadata fields that could be used as labels as well. See our project README.md for a complete description.

#### F.2.6  Is any information missing from individual instances?

We have included all relevant information from our crowdsourcing effort, and we provide links into the Yelp Academic Corpus, which contains additional metadata.

#### F.2.7  Are relationships between individual instances made explicit?

Yes.

#### F.2.8  Are there recommended data splits (e.g., training, development/validation, testing)?

Yes, we have defined a train/dev/test split, using a procedure described in Section 3.7 of our paper.

#### F.2.9  Are there any errors, sources of noise, or redundancies in the dataset?

There are likely to be errors stemming from the fact that the corpus is a naturalistic one that was processed in heuristic ways and labeled using a large-scale crowdsourcing effort. As we discover errors, we will update the dataset and associated documentation.

#### F.2.10  Is the dataset self-contained, or does it link to or otherwise rely on external resources?

The dataset is self-contained, but it can be linked with the separate Yelp Academic Dataset for additional context. Yelp controls the Yelp Academic Dataset and could stop distributing it at any time.

### F.3 Will the dataset be distributed under a copyright or other intellectual property (IP) license, and/or under applicable terms of use (ToU)?

We are distributing the dataset with a Creative Commons 4.0 International license.[4]

There are no fees associated with using our dataset. We impose no restrictions on its usage ourselves, but we are also not in a position to fully adjudicate the question of whether it inherits the terms of the Yelp Academic Dataset[5] in virtue of using text snippets from that resources. This is a complex legal question concerning the limits of fair use. We recommend that the dataset be used only for research purposes.

#### F.3.1 Does the dataset contain data that might be considered confidential?

No. All the data in Karenina is already public data taken from the Yelp site and included by Yelp in its Yelp Academic Dataset.

#### F.3.2 Does the dataset contain data that, if viewed directly, might be offensive, insulting, threatening, or might otherwise cause anxiety?

It is possible that some texts included in our dataset would be regarded as offensive. The texts are derived from online reviews, which can be very negative along many dimensions. We have not tried to identify or remove sentences that might cause offense.

#### F.3.3 Does the dataset relate to people?

Yes, it contains records of communicative acts by people, and often about people.

#### F.3.4 Does the dataset identify any subpopulations (e.g., by age, gender)?

It does not do this in any way that we are aware of, and it is not out intention to identify specific populations.

#### F.3.5 Is it possible to identify individuals (i.e., one or more natural persons), either directly or indirectly (i.e., in combination with other data) from the dataset?

Yes, there is a path to such identification, since the Yelp Academic Dataset includes identifying information about its business and users, and our dataset links into that dataset.

#### F.3.6 Does the dataset contain data that might be considered sensitive in any way (e.g., data that reveals racial or ethnic origins, sexual orientations, religious beliefs, political opinions or union memberships, or locations; financial or health data; biometric or genetic data; forms of government identification, such as social security numbers; criminal history)?

We believe the answer to be "No", except insofar as the Yelp Academic Dataset might itself contain such information.

### F.4 Collection Process

#### F.4.1 How was the data associated with each instance acquired?

The example texts were extracted from the Yelp Academic Dataset according to the procedure described in Section 3.1, and they were labeled according to the procedure described in Section 3.3 and Appendix B.

#### F.4.2 What mechanisms or procedures were used to collect the data?

All the information in the corpus was collected using Web applications.

---

[4]https://creativecommons.org/licenses/by/4.0/

[5]https://s3-media3.fl.yelpcdn.com/assets/srv0/engineering_pages/bea5c1e92bf3/assets/vendor/yelp-dataset-agreement.pdf

**F.4.3  If the dataset is a sample from a larger set, what was the sampling strategy?**

Our sampling strategy is described in Section 3.1 of our paper.

**F.4.4  Who was involved in the data collection process (e.g., students, crowdworkers, contractors) and how were they compensated (e.g., how much were crowdworkers paid)?**

The data collection was done through Amazon Mechanical Turk. The methods, including payment information, are documented in Section 3.3 and Appendix B of our paper.

**F.4.5  Over what timeframe was the data collected?**

March 2021 to June 2021. The texts in the Yelp Academic Dataset are from the period October 2004 to December 2019.

**F.4.6  Were any ethical review processes conducted?**

No. The annotation process was done using tools and techniques developed in-house by Motive Software.

**F.4.7  Did you collect the data from individuals directly, or obtain it via third parties or other sources (e.g., websites)?**

The texts were extracted from the Yelp Academic Dataset, and the labels were assigned by individual crowdworkers.

**F.4.8  Were the individuals in question notified about the data collection?**

Yes.

**F.4.9  Did the individuals in question consent to the collection and use of their data?**

We assume that participation in our tasks constitutes consent.

**F.4.10  If consent was obtained, were the consenting individuals provided with a mechanism to revoke their consent in the future or for certain uses?**

N/A, though workers were free to contact us to request that we not use their input. We have not received such requests. All our labels are assigned anonymously, and no information about crowdworkers' identities is included in our dataset.

**F.4.11  Has an analysis of the potential impact of the dataset and its use on data subjects (e.g., a data protection impact analysis) been conducted?**

We do not consider our dataset to belong to the sort of high-risk category that would require such an analysis.

**F.5  Preprocessing/cleaning/labeling**

**F.5.1  Was any preprocessing/cleaning/labeling of the data done (e.g., discretization or bucketing, tokenization, part-of-speech tagging, SIFT feature extraction, removal of instances, processing of missing values)?**

Our preprocessing steps are described in Section 3.1 of our paper.

**F.5.2  Was the "raw" data saved in addition to the preprocessed/cleaned/labeled data (e.g., to support unanticipated future uses)?**

The raw data in our case is the publicly available Yelp Academic Dataset.

### F.5.3 Is the software used to preprocess/clean/label the instances available?

We use the NLTK sentence tokenizer [4]. We also use pretrained classifier models heuristically to help with sampling. These models are not publicly available, since they belong to Motive Software.

## F.6 Uses

### F.6.1 Has the dataset been used for any tasks already?

As of this writing, it has been used only for the experiments in our paper.

### F.6.2 Is there a repository that links to any or all papers or systems that use the dataset?

No.

### F.6.3 What (other) tasks could the dataset be used for?

We are not aware of tasks outside of emotion analysis that the dataset could be used for, though it's possible that it will find uses in tasks that involve emotion analysis as a component or that would benefit from emotion labels.

### F.6.4 Is there anything about the composition of the dataset or the way it was collected and preprocessed/cleaned/labeled that might impact future uses?

Yes. The dataset is focused on consumer healthcare experiences as reported publicly on the Yelp website. This will delimit its range of useful and responsible applications.

### F.6.5 Are there tasks for which the dataset should not be used?

We feel that the dataset should be used only for emotion analysis of publicly reported consumer healthcare experiences. Applications beyond that are speculative and should be explored cautiously.

## F.7 Distribution

### F.7.1 Will the dataset be distributed to third parties outside of the entity (e.g., company, institution, organization) on behalf of which the dataset was created?

Yes, the dataset is publicly available at `https://github.com/sudhof/karenina`.

### F.7.2 How will the dataset will be distributed?

Via `https://github.com/sudhof/karenina`.

### F.7.3 When will the dataset be distributed?

It is presently available.

### F.7.4 Will the dataset be distributed under a copyright or other intellectual property (IP) license, and/or under applicable terms of use (ToU)?

We have released the dataset under a Creative Commons Attribution 4.0 International License.[6]

### F.7.5 Have any third parties imposed IP-based or other restrictions on the data associated with the instances?

No.

### F.7.6 Do any export controls or other regulatory restrictions apply to the dataset or to individual instances?

No.

---

[6]`https://creativecommons.org/licenses/by/4.0/`

### F.8 Maintenance

### F.8.1 Who is supporting/hosting/maintaining the dataset?

The creators of the dataset are supporting and maintaining it.

### F.8.2 How can the owner/curator/manager of the dataset be contacted (e.g., email address)?

Sudhof can be contacted on Github or by email.

### F.8.3 Is there an erratum?

Not at present.

### F.8.4 Will the dataset be updated (e.g., to correct labeling errors, add new instances, delete instances)?

Yes.

### F.8.5 If the dataset relates to people, are there applicable limits on the retention of the data associated with the instances (e.g., were individuals in question told that their data would be retained for a fixed period of time and then deleted)?

We are not aware of such limits.

### F.8.6 Will older versions of the dataset continue to be supported/hosted/maintained?

Yes, they will be version-controlled. The only exception is that any instances we are required to remove will be removed from archived versions as well.

### F.8.7 If others want to extend/augment/build on/contribute to the dataset, is there a mechanism for them to do so?

Yes. People can contact Sudhof via Github or by email.

