# OpenReview forum: "Karenina: Modeling the Complexity of Negative Emotions to Better Serve Industry Goals"
_NeurIPS.cc/2021/Track/Datasets_and_Benchmarks/Round1 — Submitted to NeurIPS 2021 Datasets and Benchmarks Track (Round 1)_

### Official Review · Reviewer_cwgc · 2021-06-25
**Possibly useful dataset but lacking some justifications**

**Rating:** 5
**Confidence:** 4
**Correctness:** See [weakness].

**Strengths:**

- S3.2: Label space is well-justified by linking proposed sentiment categories to actionable insights.

- The dataset is of reasonable size to train/test models, therefore it could be useful for future applications.

**Weaknesses:**

- S3.1: Some designed choices are not justified in the paper. For example, why annotate reviews on the sentence level? Would this split ignore context and affect the overall validity of the dataset?

- S3.1: Would possible fake reviews on Yelp affect the validity/usefulness of the dataset?

- S3.1: The authors write "(2) extensive patient experience data is collected due to regulations" for the healthcare industry, but I fail to see how this is related to the dataset selection since Yelp reviews are voluntarily submitted by users?

- S3.4: Because of the way the dataset was annotated, the authors are unable to measure inter-annotator agreement in the dataset, therefore the dataset is missing a measure of consistency.

- S3.5: I think the correlations among emotions are severely underestimated due to the "top 2" labeling strategy, because additional (possibly correlated) emotions are omitted if they are not "top 2", therefore the correlation measure ignores this part.

- S3.5: High correlations among emotions, plus unmeasured consistency, makes me doubt the overall quality of the dataset, e.g., the example in Table 1 "One time he made my neck worse!" seems more angry than disappointed to me...

(modified: 20 Jul 2021)

**Additional Feedback:**

None.

**Clarity:**

The paper is reasonably well-written, but I think the authors can provide more information on the Yelp dataset.

**Documentation:**

Yes.

**Ethics:**

None.

**Relation To Prior Work:**

Yes.

**Summary And Contributions:**

This paper offers a new dataset Karenina that contains finer-grained emotion labels beyond negatives and positives.

---

### Official Review · Reviewer_n5Am · 2021-06-28
**Interesting dataset, but concerns about validity and terms of use.**

**Rating:** 4
**Confidence:** 4

**Strengths:**

- Clear motivation: The paper motivates clearly why current datasets often don't fulfill industry needs and why in particular for customer support more fine-grained distinctions of negative sentiment are important.
- Relevance to the community:  I can see this dataset being of interest to a large number of people, in both industry and academia (but see comments regarding terms of use in ‘Documentation’).
- Overall the paper is clearly written and easy to follow.
- Impact statement was clear and to the point.
- Clear documentation  (e.g. datasheet, screenshots of task for workers).
- Great that the release contains the individual annotations (and not only the aggregated/inferred ones).

**Weaknesses:**

The following two weaknesses are a reason for me to recommend rejection:

- Unclear terms of use (see ‘Documentation’)
- Ad hoc data collection and labeling (see ‘Correctness’)

In addition:
- The “novelty” aspect is somewhat limited. The dataset mostly differs from existing datasets by providing fine-grained negative emotions. There are other datasets with even more fine-grained emotion labels, however these don’t focus on this particular domain. This paper doesn’t present a new task or a better way to annotate emotions.  (To me, this is not a reason to recommend rejection, but I wanted to mention it as it depends on the goals of this track.)


**Additional Feedback:**

The text could more clearly state whether the labeling/classification is at the level of reviews or sentences.

**Clarity:**

The paper is clearly written. The fact that the annotators were crowdworkers was left out of the main paper. I think this is important to state in the main paper, even though further details are provided in the appendix.


**Correctness:**

The main weakness of the paper is the somewhat ad hoc data collection and labeling approach, which makes me question the validity of the data. I'm also unsure about the exact experimental setup.

**Taxonomy is not grounded strongly in literature**

The taxonomy is mostly based on the authors’ own experience doing emotion analysis in industry (Introduction). It is not grounded strongly in relevant literature. For example, the emotion taxonomy (3.2) is presented with almost no links to literature. How do these definitions relate to definitions used in previous work on emotion (classification)?

**Number of labels**

The authors aimed for multiple labels for each comment--but crucially they don’t motivate why this is important. The most related dataset is GoEmotions. The authors argue that this dataset has as limitation that it contains relatively few examples with more than one emotion. However, they don't explain why this is problematic. What if for many comments only one emotion label is relevant? The validity of the annotations should focus on whether the annotations capture the emotions in the text, not on the number of labels. The annotation task was set up to ask each annotator to provide 2 labels for a comment. This just sounds very ad hoc to me.  The sentence “We piloted a ‘Top 1’ variant and found it to be good as well, but it yielded fewer secondary and tertiary emotions.” also raises questions.

**No annotator agreement reported**

No additional validation was done: agreement between workers was not reported. Validation could also be stronger by showing that their setup would result in better agreement with experts (for example) compared to previous annotation setups.

**Data collection**

The authors used simple heuristic classifiers to reduce the number of positive or neutral texts in the corpus. This could be explained in more detail. What types of biases could this introduce to the data distribution? What kind of "simple" classifiers were used exactly?

**Label inference**

Unclear, why 33% of the annotators? It's also unclear why the number of annotators is different per instance.

**Multi-class vs. multi label**

In this dataset, each comment can have multiple labels. A multi-label classification approach therefore seems natural. The authors  however used multi-class classifiers. I'm unsure about the exact implementation/setup. The sentence  "For F1 scores, where there are multiple top labels for an example, we duplicate the example with each of those labels." is unclear.


**Documentation:**

I have concerns about the terms of use for this dataset. This dataset is based on an existing dataset, the Yelp Academic Dataset, and it’s unclear how the terms of use of this dataset relate to the Yelp Academic Dataset.

*"We impose no restrictions on its usage ourselves, but we are also not in a position to adjudicate the question of whether it inherits the terms of the Yelp Academic Dataset in virtue of using text snippets from that resources."* (Appendix)

This seems problematic. It’s unclear whether the authors actually tried to contact Yelp about this to help clarify this. Even though the authors state *“The authors bear all responsibility in case of violation of rights.”* this seems important to address before releasing the data.

Furthermore, Yelp mentions that everyone needs to sign their terms of use (https://www.yelp.com/dataset/documentation/faq), yet this dataset distributes the raw text. And while this new dataset is specifically designed with industry purposes in mind, the Yelp website states: *"[..] However, this falls under the commercial use category, and therefore you will have to first contact us by email at dataset@yelp.com to request permission."*


**Ethics:**

The following seems problematic, because it's below minimum wage (at least in the US) *"The estimated hourly wage paid to participants was US$5.25"* (appendix B)

**Relation To Prior Work:**

The paper motivates clearly why there is a need for a dataset like this. The used emotion taxonomy could be more clearly grounded in and compared to related work.

In general, the related work section on emotion analysis could be stronger (there's a lot of work in NLP, for example I'm thinking of work by Saif M. Mohammad).

**Summary And Contributions:**

Thank you for the author response and the clarifications (incl. the clarification about the wages paid).
Two main concerns remain: the justification behind the annotation process (e.g. the top 2 setup), and the experimental setup.
Given that the paper's main contribution is a multi-label setup, I still find it confusing that in the experiments a multi-class setup is used (with as gold label the label with the highest count), instead of modeling it as a multi-label problem.

====

Although there are already many sentiment analysis datasets available, this paper argues that current datasets do not suit industry purposes well as there is a need for more fine-grained distinctions of negative sentiment.
This paper therefore presents Karenina, a labeled dataset of 25k consumer healthcare experience comments. This dataset contains labels for six negative emotions, and many comments have more than one label.


Contributions:
- A dataset of 25k consumer healthcare experience comments. It differs from most datasets by having 1) labels for six negative emotions (+neutral and positive), 2) multiple labels per comment.
- Baseline experiments (random classifier, two BERT based classifiers)
- Exploratory analysis showing how this dataset could be used to gain more insights (e.g. finding terms and associated emotions).

---

### Official Review · Reviewer_KraU · 2021-07-06
**solid dataset on fine-grained negative emotions**

**Rating:** 5
**Confidence:** 4

**Strengths:**

1. Large, carefully curated dataset focusing on negative emotions. Key attributes of this dataset are the rigor with which it was collected, the focus on the negative sentiment which will surely be useful, and a focus on reviews in health applications.
2. Good idea to introduce a new weighted version of standard F1 metrics related to closeness of different mistakes, although this is a relatively straightforward and intuitive contribution.
3. Nice case study at the end of the work, showing different emotional profiles for different keywords


**Weaknesses:**

There exist numerous datasets on sentiment analysis, and as the authors pointed out, especially GoEmotions, which have pursued similar tasks of fine-grained emotion annotation. GoEmotions contains more (i.e. 58k v 25k) annotated comments, from a larger range of emotions (i.e. 27 v  8). Between these two datasets, 3 emotions overlap (anger, disappointment, confused), although other classes are very similar (optimism or approval vs happy, etc). Therefore, in terms of dataset size and breadth this is not groundbreaking. However, it is noted that the current dataset does include multiple labels, which is new.

The idea of correlated emotions is also present in GoEmotions, so this sort of analysis is not new, and similar qualitative findings were observed.

The modeling is relatively standard - taking from BERT, either in the form of a binary or multi-class classifier.

Extended comparison between models trained on Karenina vs GoEmotion would be useful, to help show that Karenina is more uniquely suited for a particular challenge. Otherwise, I think the average person tackling sentiment classification, whether in healthcare consumer feedback or other arenas, may start with GoEmotions.


**Additional Feedback:**

A solid paper presenting a dataset for a well-studied task, that will certainly be used, but perhaps not novel enough in terms of task or analytical metrics, for NeurIPS.

**Clarity:**

Paper is very well and clearly written. Supporting information helps to clear up anything not explicitly described in the main text.

**Correctness:**

Claims seem correct, and I believe the dataset, and its method of collection, are sound.

**Documentation:**

Yes, in the main text, supporting information, and repo, there is significant details on dataset collection, organization, availability, maintenance, as well as ethical and responsible use.

**Ethics:**

The authors mention ethnical concerns related to emotion analysis and indeed bring it to the readers attention.

**Relation To Prior Work:**

Previous work in sentiment analysis and emotional analysis benchmarks is clearly described in the main text. Importantly, they describe their contribution as most similar to GoEmotions, with the distinctions being a specialized healthcare domain and using multiple labels (although some part of GoEmotions also uses multiple labels).

**Summary And Contributions:**

The authors present a large, and carefully crafted dataset which will be very useful for modeling negative emotions in a more fine-grained way. The authors present a very detailed account for the dataset collection, and rigorously define labels through agreement of atleast 4 reviewers. Additionally, the authors present an interesting discussion and correlation between different emotional labels (i.e. the strong relationship between stressed and worried), and present a reasonable framework / loss for when the model is not exactly right, but still 'rightish', which makes sense. The researchers present a standard set of models as baselines on this new task, but show that the simplest binary classification model work best.

---

### Decision · Program_Chairs · 2021-07-26

**Decision:**

Reject

**Comment:**

This paper presents a new sentiment analysis dataset that extends the standard formulation of the task to include a broader range of labels, specifically different negative emotion categories.

Reviewers agree that the label space presented in this dataset is well motivated and offers a useful contribution. However, the reviewers are in agreement that this paper does not meet the bar for publication at this time. There are some concerns regarding the documentation of the dataset and terms of use, all of which questions the overall validity of the benchmark.

I encourage the authors to revise this work in light of the reviews and submit to the second round of this track.